# Analysis of Land Use and Land Cover Changes through the Lens of SDGs in Semarang, Indonesia

**Mira Kelly-Fair** [1,*], **Sucharita Gopal** [1,*], **Magaly Koch** [1], **Hermin Pancasakti Kusumaningrum** [2], **Muhammad Helmi** [2], **Dinda Khairunnisa** [2] **and Les Kaufman** [3]

[1] Department of Earth and Environment, Boston University, Boston, MA 02215, USA; mkoch@bu.edu
[2] Center for Coastal Rehabilitation and Disaster Mitigation Studies–CoREM, Diponegoro University, Semarang 50275, Indonesia; herminpk@live.undip.ac.id (H.P.K.); muhammadhelmi69@gmail.com (M.H.); khairunnisa.dinda29@gmail.com (D.K.)
[3] Department of Biology, Boston University, Boston, MA 02215, USA; lesk@bu.edu
[*] Correspondence: mirakf@bu.edu (M.K.-F.); suchi@bu.edu (S.G.)

**Abstract:** Land Use and Land Cover Changes (LULCC) are occurring rapidly around the globe, particularly in developing island nations. We use the lens of the United Nations' Sustainable Development Goals (SDG) to determine potential policies to address LULCC due to increasing population, suburbia, and rubber plantations in Semarang, Indonesia between 2006 and 2015. Using remote sensing, overlay analysis, optimized hot spot analysis, expert validation, and Continuous Change Detection and Classification, we found that there was a spread of urban landscapes towards the southern and western portions of Semarang that had previously been occupied by forests, plantations, agriculture, and aquaculture. We also witnessed a transition in farming from agriculture to rubber plantations, a cash crop. The implications of this study show that these geospatial analyses and big data can be used to characterize the SDGs, the complex interplay of these goals, and potentially alleviate some of the conflicts between disparate SDGs. We recommend certain policies that can assist in preserving the terrestrial ecosystem of Semarang (SDG 15) while creating a sustainable city (SDG 11, SDG 9) and providing sufficient work for individuals (SDG 1) in a growing economy (SDG 8) while simultaneously maintaining a sufficient food supply (SDG 2).

**Keywords:** SDGs; remote sensing; climate change; land cover; geographic information systems; sustainability





## 1. Introduction

Global land cover is impacted by changing natural environmental conditions and anthropogenic activities. Differentiating changes in land cover due to natural variability compared to anthropogenic activities requires understanding how natural and human systems interact [1] and distinguishing between land cover and land use. Land cover denotes the physical and biotic character of the land surface [2]. Examples of land cover include forests, grasslands, wetlands, and water. Land use involves human activities transforming the land cover, such as office parks and golf courses [3]. Land Use and Land Cover Changes (LULCC) are occurring worldwide, driven mainly by anthropogenic activities [4]. LULCC considerably alters the earth's biophysical, biogeochemical, and energy exchange processes [5–8]. LULCC caused by urban expansion and deforestation can affect the global carbon budget and have long-term climatic impacts [9–11].

LULCC impacts range from local to global scales [12]. LULCC has resulted in deforestation and agricultural expansion, as well as urbanization. Urbanization often accompanies deforestation in the tropics, as evidenced in the Brazilian Amazon and elsewhere [8,13]. LULCC is often associated with agricultural expansion [14–16] in developing countries. LULCC results from engineering projects [17,18] and often involves changes in infrastructure, such as the construction of hydroelectric dams [19], impacting biodiversity [20,21]

and settlements of the indigenous population [22]. Hence, assessing the consequences begins with understanding what, where, and when LULCC is occurring, its drivers, and its impacts in the present and future.

With limited land area and rich biodiversity, the Indonesian archipelago is experiencing LULCC driven by climate change and anthropogenic drivers. Climate change in this area is projected to lead to increased temperatures, intense rainfall [23], and sea-level rise [24]. Sea-level rise, combined with land subsidence, results in increased flooding, increased storm surges, shoreline erosion, and storm exposure in coastal areas [25,26]. This scenario is further accelerated by anthropogenic forces such as groundwater extraction and land development [27] in coastal regions, including Semarang. Such changes have led to the capital moving from Jakarta to the newly established city of Nusantara, in Borneo's East Kalimantan province [28].

LULCC in Indonesia over the last 20 years has been driven by anthropogenic drivers, including increasing population, rural to urban migration, and economic development resulting in denser and larger urban centers. This country has the world's fourth-largest population (around 273 million in 2020) [29]. Its urban population has steadily increased from 50% in 2010 to 57% in 2020 [30] and may grow to 67% by 2050 [31]. Its urban land area is increasing at a rate of 1.1% each year, driven by high population density and growth [29]. Accompanying this urbanization are societal transformations marked by patterns of affluence and poverty. A World Bank report [32] notes that the middle class has grown faster than other groups with substantial consumption increases. This urban demographic transition is associated with increased greenhouse gas emissions due to greater consumption of resources, particularly fossil fuels [32,33].

In Indonesia, the urban areas are economically, socially, and environmentally coupled with rural agricultural and forested areas. McGee [34,35] coined the term desakota to describe places in the extended periphery of large cities, in which urban and agricultural patterns of land use co-exist and are intensively intermingled. The term desakota comes from the Indonesian word Desa, meaning "village," and Kota, meaning "city" [34]. Using the desakota framework, the housing and infrastructure needs of the growing urban centers must be balanced with the land needed for agriculture and food production at the periphery. Indonesian commercial agriculture focuses on cash crops such as rubber and palm oil in its forested landscapes. These cash crops provide both agricultural and non-agricultural employment opportunities in rural and semi-urban areas [36].

Protecting this forested landscape as biodiversity habitats is also essential to mitigating climate change, as deforestation and forest degradation represent a significant source of greenhouse gas emissions [37]. Thus, urbanization and economic development have led to LULCC marked by urban expansion, commercial agriculture, deforestation, and decreasing biodiversity. To tackle the challenges posed by such LULCC, it is imperative to have an effective system of urban planning and governance, particularly in Indonesia. We will be examining sustainable development and LULCC in the city of Semarang in Java, Indonesia (Figure 1), the focus of this paper.

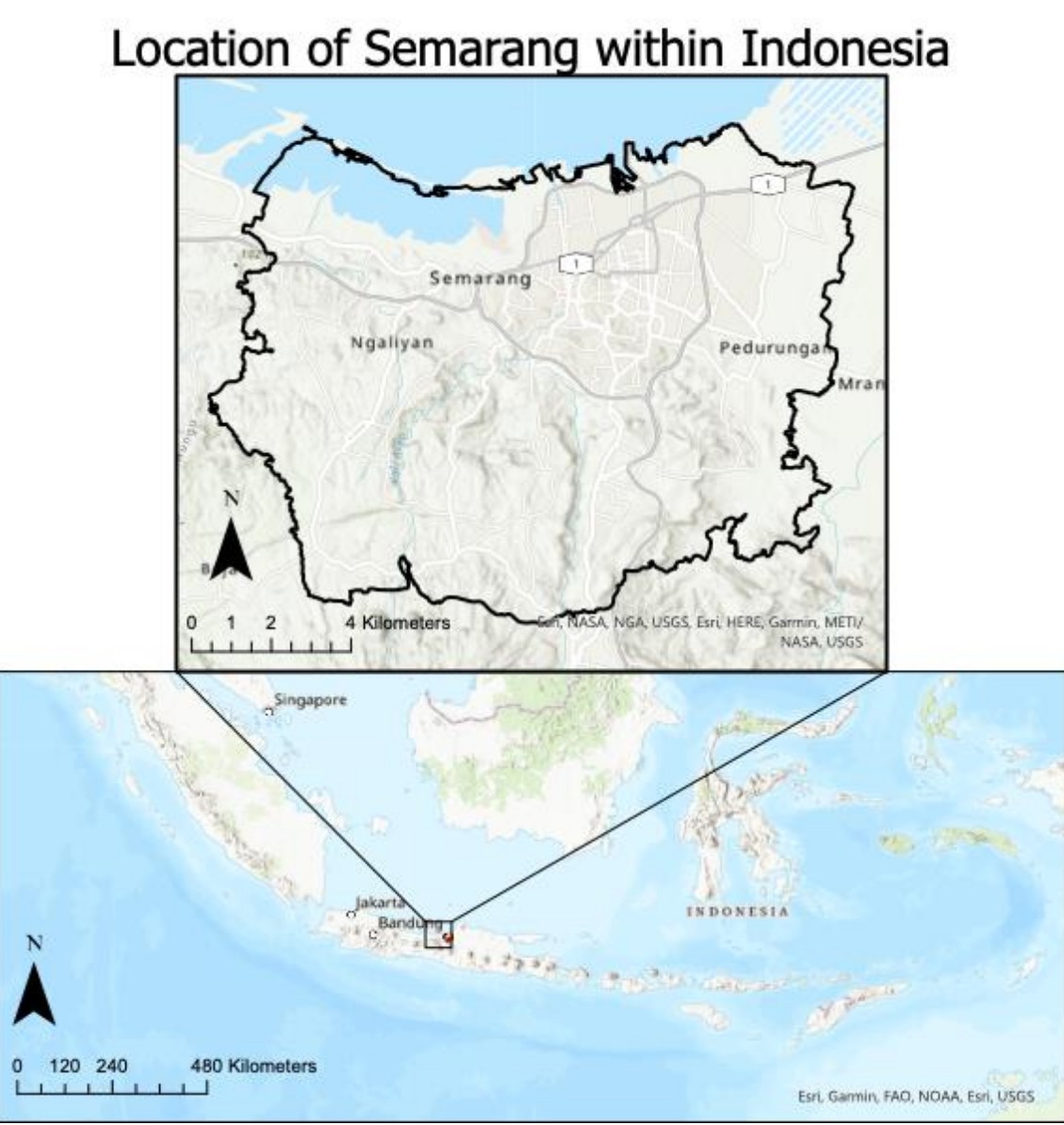

**Figure 1.** This inlay shows the outline of Semarang City within the context of both the island of Java and the country of Indonesia. This country is located between the Indian and Pacific Oceans in Southeast Asia.

### 1.1. Sustainable Developement Goals

The United Nations established Sustainable Development Goals (SDG) in September 2015 to respond to increasing population and consumption, focusing on different aspects of sustainability prevalent in developing communities around the globe [38]. This UN SDG global indicator framework consists of 244 indicators across 17 goals and 169 targets. Six percent of all indicators are for Goal 11 (Sustainable cities and communities). The official mission of SDG 11 is to "Make cities inclusive, safe, resilient and sustainable" [38].

With its limited landmass and burgeoning population, Indonesia has to consider the suitable SDGs to simultaneously "take urgent and significant action to reduce degradation of natural habitats, halt the loss of biodiversity" (15.5; [38]). At the same time, it has to satisfy the somewhat conflicting goals calling to "sustain per capita economic growth…at least 7 percent gross domestic product growth" (8.1; [38]) and "develop…infrastructure to support economic development and human well-being" (9.1; [38]). As stated previously, SDG 11 targets sustainable and resilient growth in cities. Goal 11.1 aims to "Support positive economic, social and environmental links between urban, peri-urban and rural

areas by strengthening national and regional development planning" (11.1; [38]). Indonesia needs to undertake climate action, SDG 13, in its urban planning of coastal cities. Goal 13.1 calls for strengthening resilience and adaptive capacity to deal with climate-related hazards and natural disasters [38]. Although economic growth and infrastructure goals may align, they are at odds with reducing degradation and deforestation.

Prior studies on SDG in the Indonesian urban context draw attention to challenges in urban governance [39], food security [40], and bioenergy development [41]. The major objective of this study is to quantitatively evaluate the impact of LULCC under rapid urbanization drawing on the UN SDG framework. Based on a case study in Semarang, Indonesia, we discuss the LULCC in a recent period of urban expansion. This approach is employed to address the following questions: How does the LULCC result in competing SDGs—protecting biodiversity, ensuring food security and livelihoods, and sustainable urban growth? How do we measure such changes and understand the tradeoffs in various regions of the city of Semarang? In this paper, we develop a relevant methodology to analyze the LULCC that characterizes the urban growth in Semarang using geospatial data, both GIS and remote sensing. Our study fills a knowledge gap in linking LULCC studies to UN SDGs in the heavily urbanized area, Semarang, in Java.

### 1.2. Study Area: Semarang City in Indonesia

Semarang City is the seventh-largest city in Indonesia [42]. In 2006, Semarang had a population of 1.5 million, which increased to approximately 1.7 million by 2015, a growth rate of almost 22 thousand people per year [42]. In addition to population growth, the Indonesian Gross National Income (GNI) per capita more than doubled during that same period from $315.5 billion USD in 2006 to $886.55 billion USD in 2015 [43]. Semarang is located on the northern coast of Java, which is prone to rising sea levels and land subsidence, and thus a bellwether case study for LULCC and the SDGs. Understanding why Semarang City is experiencing different patterns of LULCC is essential to understanding the risks this city may face in the future and the consequences of meeting the UN SDG goals. Not only are there more people in Semarang, but they are also more affluent than ever before, according to a recent World Bank report [43]. This new wealth and greater population can lead to an increase in infrastructure as well as increased demands for food and economic opportunity [44]. It is critical to understand where and why LULCC are happening throughout the city, not only in terms of demographics but also as climate change and tectonic activities result in increasing sea levels and land subsidence [44–53]. This study aims to ensure that Semarang City develops and urbanizes using a sustainable urban planning framework to meet UN SDG 11. This framework can inform policy decisions to guarantee the safety of its people and reduce exposure to climate change (SDG 13.1) and other natural disasters [38,44]. By looking at these factors, we hope to determine if Semarang City was developing sustainably before implementing the UN SDGs in 2015.

### 1.3. Geospatial Data for Monitoring LULCC

Remote sensing instruments are the most viable option for data-driven mapping, monitoring, and assessment of the long-term, large-scale status and dynamics of LULCC. Landsat satellites have continuously acquired images of Earth's surface since 1972. Now containing more than forty years of multi-spectral reflectance observations, the Landsat data archive represents an unparalleled record of change on Earth's surface. The spatial (30 m) and temporal (8–16 day) resolution of Landsat make it a unique resource for anyone interested in the status and dynamics of land cover at local and regional scales [54–57]. With the entire sequence of Landsat imagery now freely available through the USGS delivered via Google Earth Engine, we can address questions related to the dynamics of LULCC in Semarang.

## 2. Materials and Methods

For this LULCC study, we used geospatial data including remote sensing and GIS data. The first set of remote sensing data is helpful in uncovering LULCC from 2006 and 2015. The second set of GIS data provided us with town planning details. We also consulted experts in Semarang to validate our LULCC findings.

As shown in Figure 2, we processed two images of land use and land cover (LULC) of Semarang collected in 2006 and 2015. This World Bank data product [58] contains spatially explicit information on different land use and land cover in the City of Semarang. The data product makes a distinction between the large urban and core urban areas of the city. The large urban class is aggregated at Level 1 or 2 using Landsat-5 and Sentinel-2 data. On the other hand, the core area has finer details aggregated at levels 3 or 4 using high resolution Quickbird-2 data. The levels are defined and based off of the hierarchical system used by the European Commission's Urban Atlas [59]. Each is sourced from different instruments and the whole data product is available for download from the World Bank website. We utilized these data since they are processed and contain the right level of detail for the purpose of this study.

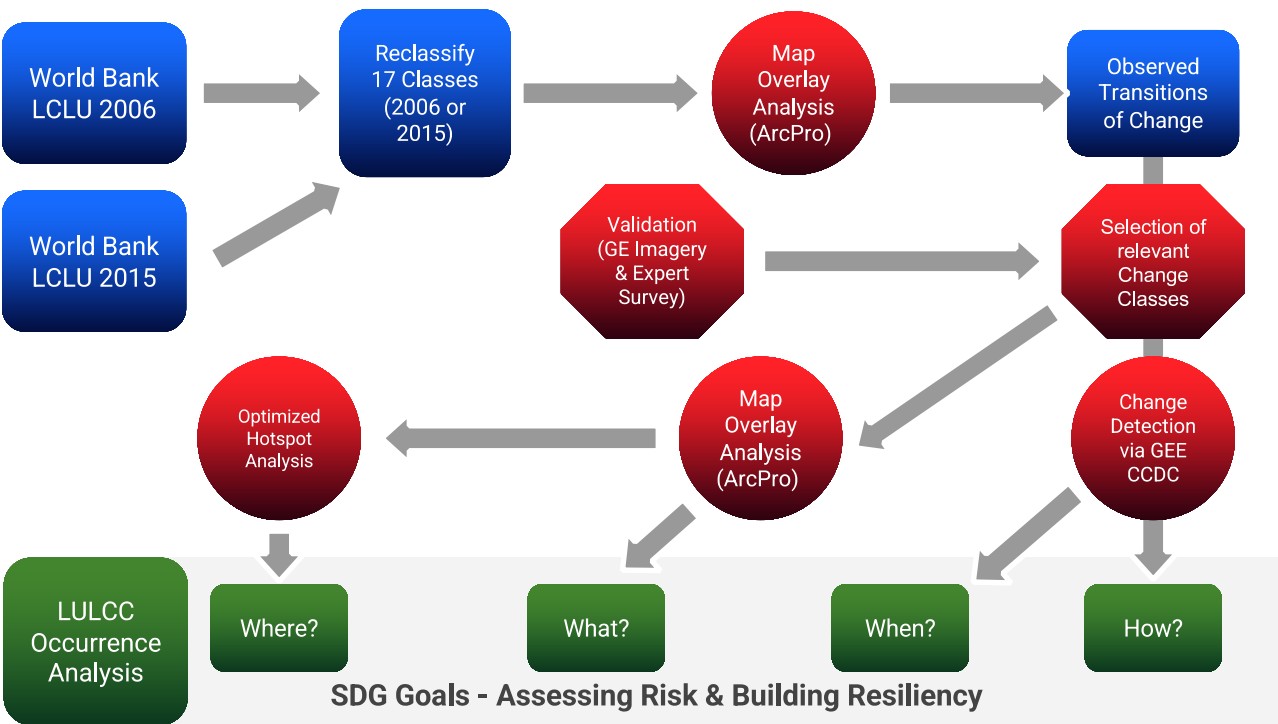

**Figure 2.** Workflow for Study; World Bank LULC data from 2005 and 2016 was reclassified into 17 classes. Figure shows data flow and algorithms utilized for transforming World Bank LULC data from 2005 and 2016.

We want to ask four fundamental questions in change analysis: what, where, when and how. We detail the analysis undertaken to tackle each component of change shown in Figure 2. As a first step, we utilized two World Bank LULC maps of Semarang City, Indonesia created by the Earth Observation for Sustainable Development (eo4sd) program of the European Space Agency (ESA) for 2006 and 2015 [58] (Figure 2). The 2006 data were collected using Landsat-5, and the 2015 data were collected using Sentinel-2 [58]. The raw data from ESA were divided into several different subsets ranging from general groups with 5 classes to the most specified LULC with 22 classes. We initially chose to focus on 17 classes. We processed the 17 classes of LULC datasets using ArcGIS Pro and then conducted reclassification to identify 6 classes in the two LCLU datasets in 2006 and 2015.

With the data thus prepared, we were ready to start answering our questions. The purpose of the first question was to determine what changes were occurring. To do this, we investigated change in LULC using a map overlay analysis utilizing the two images. We identified areas in Semarang that had undergone changes. From these results, we selected relevant change classes. We used ArcGIS Pro and published statistics. Note that ArcGIS Pro analysis could also be completed in QGIS or other similar GIS software. We next identified change classes. We generated a second Map Overlay to determine what LULCC occurred. For this analysis, we created an overlay of all LULC in 2006 and 2015 and then removed all parcels where the LULC did not change between those periods to focus on areas of change. This analysis addresses the question of what is changing.

Next, we wanted to determine where these changes were occurring. To do so, we used the change map from the prior steps to create a hotspot map of change. The optimized hot spot analysis uses Getis-Ord Gi* to determine parcels that are significantly clustered near other parcels with a similar attribute—in this case, the change in LULC. This map shows areas of high change and areas with little change. This subset of analysis is incredibly useful in successful urban planning.

Finally, we wanted to uncover when and how LULCC was occurring by means of the Continuous Change Detection and Classification (CCDC). Using Google Earth Engine (GEE) with code modified from others' research [54,60] (see "Data Availability" section), we applied CCDC to our processed data product. As shown in Figure 2, the CCDC enables us to answer questions related to how and when change happened since Google Earth Engine has access to all available Landsat images between 2006 and 2015. We utilized the CCDC to specifically understand the change trajectory of individual pixels—for example, agriculture converted to airport, barren land converted to agriculture, or forest converted to commercial plantations.

After conducting our analysis, we validated our results using satellite imagery and Google Street View, and expert validation. Our Semarang-based team was vital in providing expert validation and ground truthing. Due to the COVID-19 pandemic, the entire team could not be present in Indonesia, which made this communication of information even more essential. This was especially critical in instances where the satellite imagery or Google Street View was not clear in differentiating classes such as in the case of non-plantation agriculture and forest plantation. In this case, both appear similar using Landsat imagery, and Google Street View is not available in those sections. Our Semarang team was able to assist in these instances.

## 2.1. Classifying LULCC

We processed the LULCC datasets and reclassified to identify 17 classes in the two LCLU datasets in 2006 and 2015. However, the original number of classes for change analysis (128, at Level 3) was not feasible, due to the extent of analysis and validation methods using Google Earth imagery [58]. Hence, we used the six classes shown in Figure 3: agriculture, forest, commercial/industrial, urban, wetlands, and other. Figure 3 shows the land cover of the districts of Semarang City including Central Semarang, Tembalang, and Mijen. Agriculture and forest areas are prominent on the periphery of the city in the south and west, highlighting the desakota framework.

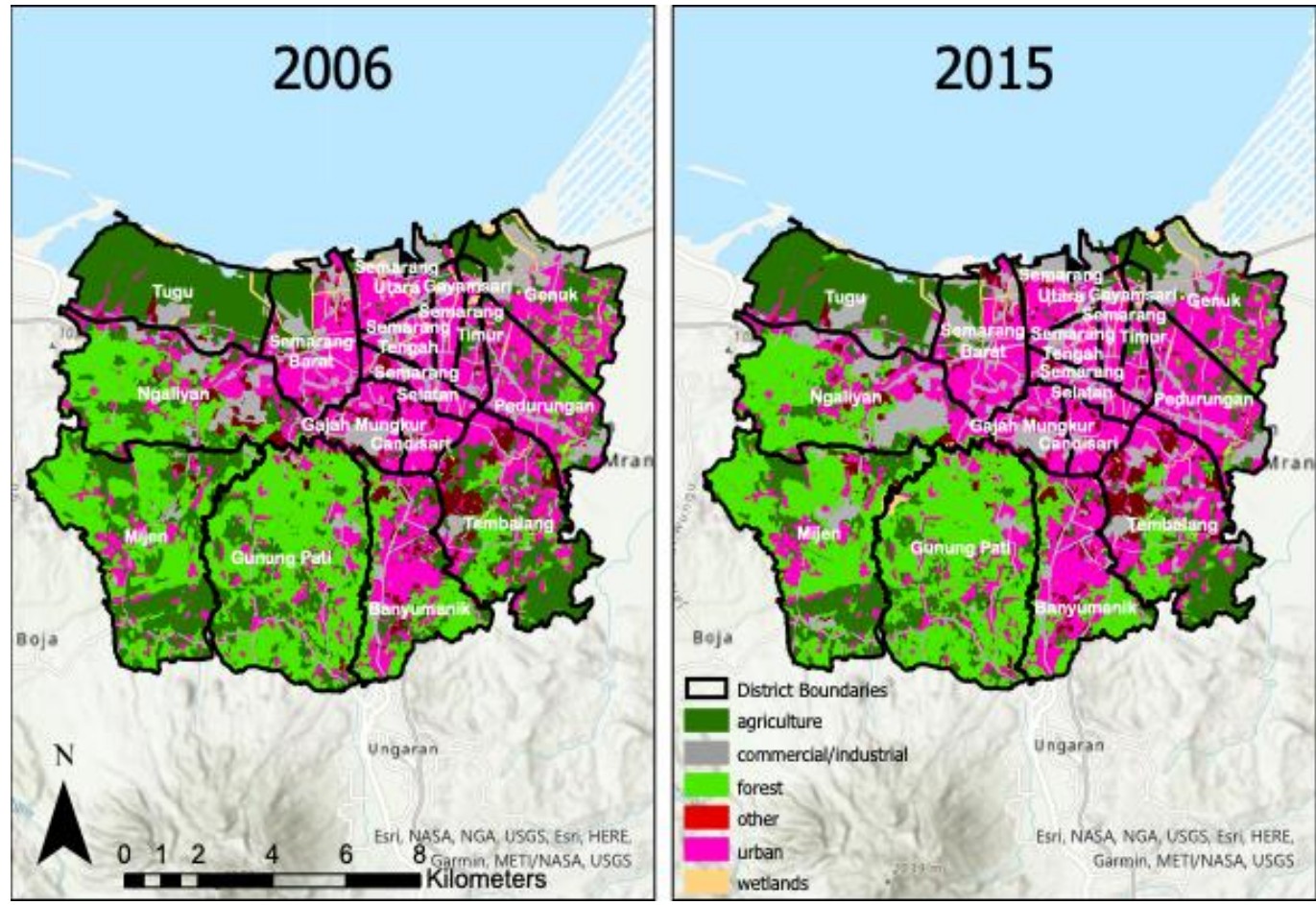

**Figure 3.** LULC in study area, 2006 and 2015, respectively.

*2.2. Identifying Significant Change Classes Using Map Overlay*

To familiarize ourselves with the data and understand the changes that occurred in Semarang City, we conducted a map overlay analysis. The goal of this overlay analysis was to determine the landcover change between 2006 and 2015 in the 17 selected classes. The results of the change matrix created from this overlay analysis helped to determine the reclassification we created (Table 1). This resulting number of 272 change classes between 2006 and 2015 could not be easily interpreted. Moreover, some change classes are too small (such as forest to wetland) and thus hard to validate. In the next step, we eliminated some LULCC classes to derive more meaningful interpretation interest.

**Table 1.** Matrix of Changes by NL_3 Classification in km$^2$; bolded are changes larger than 1 km$^2$; columns reference class in 2015 and rows represent class in 2006.

| 2015 → / 2006 ↓ | Agriculture | Airport | Cemetery | Constru-ction | Continuous Urban | Discontinuous Urban | Forest | Green Urban Area | Industrial | Inland Water | Land without Use | Mining/ Dumping | Other | Port | Recreation | Roads/Trails | Wetlands |
|---|---|---|---|---|---|---|---|---|---|---|---|---|---|---|---|---|---|
| agriculture | null | 0.841 | 0.004 | **2.683** | 0.304 | **8.319** | **10.092** | 0.774 | **3.604** | **1.485** | **1.456** | 0.495 | **1.509** | 0.000 | 0.076 | 0.133 | 0.090 |
| airport | 0.000 | null | 0.000 | 0.000 | 0.000 | 0.000 | 0.000 | 0.000 | 0.000 | 0.000 | 0.000 | 0.000 | 0.000 | 0.000 | 0.000 | 0.000 | 0.000 |
| cemetery | 0.000 | 0.000 | null | 0.000 | 0.000 | 0.540 | 0.000 | 0.013 | 0.000 | 0.000 | 0.006 | 0.000 | 0.000 | 0.000 | 0.000 | 0.001 | 0.000 |
| construction | 0.420 | 0.000 | 0.000 | null | 0.128 | **1.606** | 0.057 | 0.070 | **1.483** | 0.000 | 0.037 | 0.000 | 0.000 | 0.018 | 0.050 | 0.001 | 0.023 |
| continuous urban | 0.000 | 0.000 | 0.000 | 0.000 | null | **1.920** | 0.000 | 0.006 | 0.000 | 0.000 | 0.005 | 0.000 | 0.000 | 0.000 | 0.000 | 0.003 | 0.000 |
| discontinuous urban | 0.000 | 0.000 | 0.000 | 0.007 | 5.131 | null | 0.041 | 0.000 | 0.090 | 0.000 | 0.010 | 0.000 | 0.000 | 0.000 | 0.000 | 0.054 | 0.000 |
| forest | **2.298** | 0.000 | 0.000 | **1.108** | 0.028 | 7.809 | null | 0.370 | **1.150** | 0.142 | 0.475 | 0.075 | 0.427 | 0.000 | 0.008 | 0.205 | 0.003 |
| green urban area | 0.018 | 0.000 | 0.000 | 0.108 | 0.049 | 0.739 | 0.000 | null | 0.294 | 0.101 | 0.014 | 0.004 | 0.093 | 0.000 | 0.000 | 0.025 | 0.000 |
| industrial | 0.052 | 0.000 | 0.000 | 0.126 | 0.000 | 0.000 | 0.019 | 0.049 | null | 0.000 | 0.000 | 0.000 | 0.010 | 0.000 | 0.000 | 0.011 | 0.000 |
| inland water | 0.151 | 0.000 | 0.000 | 0.021 | 0.000 | 0.000 | 0.036 | 0.006 | 0.041 | null | 0.003 | 0.000 | 0.004 | 0.006 | 0.000 | 0.003 | 0.024 |
| land without use | 0.006 | 0.000 | 0.000 | 0.128 | 0.007 | 0.100 | 0.114 | 0.148 | 0.453 | 0.000 | null | 0.000 | 0.269 | 0.023 | 0.007 | 0.000 | 0.000 |
| mining/ dumping | 0.000 | 0.000 | 0.000 | 0.036 | 0.000 | 0.046 | 0.004 | 0.007 | 0.000 | 0.000 | 0.055 | null | 0.000 | 0.000 | 0.000 | 0.000 | 0.000 |
| other | 0.083 | 0.000 | 0.000 | 0.259 | 0.018 | 0.758 | 0.503 | 0.013 | **1.631** | 0.012 | 0.305 | 0.128 | null | 0.000 | 0.019 | 0.036 | 0.000 |
| port | 0.000 | 0.000 | 0.000 | 0.000 | 0.000 | 0.000 | 0.000 | 0.000 | 0.000 | 0.022 | 0.000 | 0.000 | 0.000 | null | 0.000 | 0.000 | 0.000 |
| recreation | 0.000 | 0.000 | 0.000 | 0.000 | 0.000 | 0.000 | 0.000 | 0.057 | 0.033 | 0.000 | 0.000 | 0.000 | 0.005 | 0.000 | null | 0.000 | 0.000 |
| roads/trails | 0.000 | 0.000 | 0.000 | 0.012 | 0.000 | 0.005 | 0.002 | 0.000 | 0.000 | 0.000 | 0.000 | 0.000 | 0.004 | 0.000 | 0.000 | null | 0.000 |
| wetlands | 0.174 | 0.000 | 0.000 | 0.026 | 0.000 | 0.055 | 0.000 | 0.004 | 0.676 | 0.148 | 0.061 | 0.000 | 0.043 | 0.014 | 0.000 | 0.008 | null |

### 2.3. Filtering LULCC Classes Using Map Overlay

We finally selected six classes, or 23 change classes, which better explained changes that occurred in Semarang over this period of time (Table 2). We conducted a second overlay analysis of this data subset and created another change matrix. We ran two different accuracy assessments of these data: the first based on a traditional validation utilizing Google Earth Pro imagery, and the second based on photography and expert validation of the changing landscape of Semarang City. This overlay analysis helped in analyzing which changes had occurred; however, further analysis was needed to determine where these changes were occurring. In the next step in our methodology, we utilized an optimized hotspot analysis.

**Table 2.** Reclassification of Simplified Classes.

| Original Class | Reclassification |
| --- | --- |
| agriculture | agriculture |
| forest | forest |
| cemetery<br>green urban area<br>other<br>recreation | other(natural/semi-natural) |
| inland water<br>wetland | wetland |
| continuous urban<br>discontinuous urban | urban |
| construction<br>airport<br>industrial<br>land without use<br>mining/dumping<br>port<br>roads/rails | industrial/commercial |

### 2.4. Optimized Hotspot Analysis

We used an optimized hotspot analysis to characterize areas that are more likely or less likely to experience changes due to their location and whether or not their neighbors experienced change. This analysis used the original ESA dataset and not just the areas which experienced change (see Sections 2.1 and 2.3). We created a binary classification where zero indicated an area with no change between 2006 and 2015 and one indicated that a change had occurred as defined in our reclassification process.

### 2.5. Continuous Change Detection and Classification (CCDC)

This change analysis helped explain what had changed in 2006 and 2015 individually but not what had changed between 2006 and 2015. To answer this question, we utilized the CCDC implemented [54,60] now on Google Earth Engine (GEE). The CCDC algorithm targets spatiotemporal change with a focus on abrupt change at the pixel scale [54,60]. To detect pixel-level changes, the approach uses a series of Fourier models to both detect large changes in the spectral reflectance of a pixel and model surface reflectance of stable segments. The general surface reflectance model follows the form:

$$Pred(x_i) = a_{0,i} + b_{0,i}x_i + a_{1,i}\cos\left(\frac{2\pi}{T}x_i\right) + b_{1,i}\sin\left(\frac{2\pi}{T}x_i\right)\ldots + a_{n,i}\cos\left(\frac{2\pi}{\frac{1}{n}T}x_i\right) + b_{n,i}\sin\left(\frac{2\pi}{\frac{1}{n}T}x_i\right)$$

where $x$: day-of-year; $i$: $i$-th band of Landsat data; $a_0$: overall surface reflectance; $b_0$: overall trend of surface reflectance; $a_1$, $b_1$: annual changes of surface reflectance; and $a_n$, $b_n$: inter-annual change in surface reflectance.

The harmonic model captures seasonal trends in reflectance with the amplitude of oscillations and model fit to observations providing information on the type of land cover, in this case due to urbanization and LULCC. The model also captures gradual change, expressed as the slope of the model. In this case, growing trees decrease red reflectance over time. Finally, the model captures abrupt shifts in land cover with the time of the shift marked with a red circle. For abrupt change detection, the model is used to predict surface reflectance values, and if observed values differ more than a defined number of standard deviations from the model predictions more often than a certain number of times in the series, the model marks a point of rapid land cover change.

In Figure 4, we can see these CCDC changes. This pixel is from an area that was converted from traditional agriculture (orange) which was replaced with young rubber plantations at the end of 2006 (green). These trees reached maturity in 2014 (purple) and then were cut at the end of 2015 (blue). We did not see the cut trees in our data as this extended slightly past the GIS dataset. However, this more thoroughly informs all the changes that happened between the years of the World Bank data.

**Figure 4.** Sample CCDC pixel of an area in Semarang that transitioned from agriculture to forest.

Additionally, we took further steps to validate the CCDC findings. First, we gathered data on the change in area cover of permeable surfaces in Semarang. To do this, we reclassified agriculture, forest, other, and wetland as permeable, and urban and commercial/industrial as impermeable. Finally, we used Google Earth Pro and Google Maps Street View with their time features to determine if rubber plantations were being run as rubber monocultures or rubber agroforestry (RAF) and to validate our Google Earth Engine results for when they were planted. Because of the desakota layout of Semarang, it is important to not only analyze Central Semarang but also districts near the periphery that are experiencing change. As such, although our results cover the entirety of Semarang, our discussion includes examples of disparate changes occurring on a large scale in several districts.

## 3. Results

Areas that experienced change between 2006 and 2015 ranged throughout Semarang City, Indonesia as seen from the optimized hotspot analysis (Figure 5). There were almost no changes in the already developed districts of Semarang Tengah (Central Semarang), Semarang Selatan (South Semarang), Semarang Timur (East Semarang), Candisari, and Gajah Mungkur (Figure 5). The few changes in these areas changed from a class called Other to Urban or from Other to Commercial/Industrial. Upon inspection, these "other" areas in question were recreational or just general green areas in what appeared to be wealthier neighborhoods. Outside of these areas, there are quite a few changes over this

decadal period of observation. Between 2006 and 2015, 55.58 km$^2$ changed their classes, approximately 14.4% of the entire area of Semarang City.

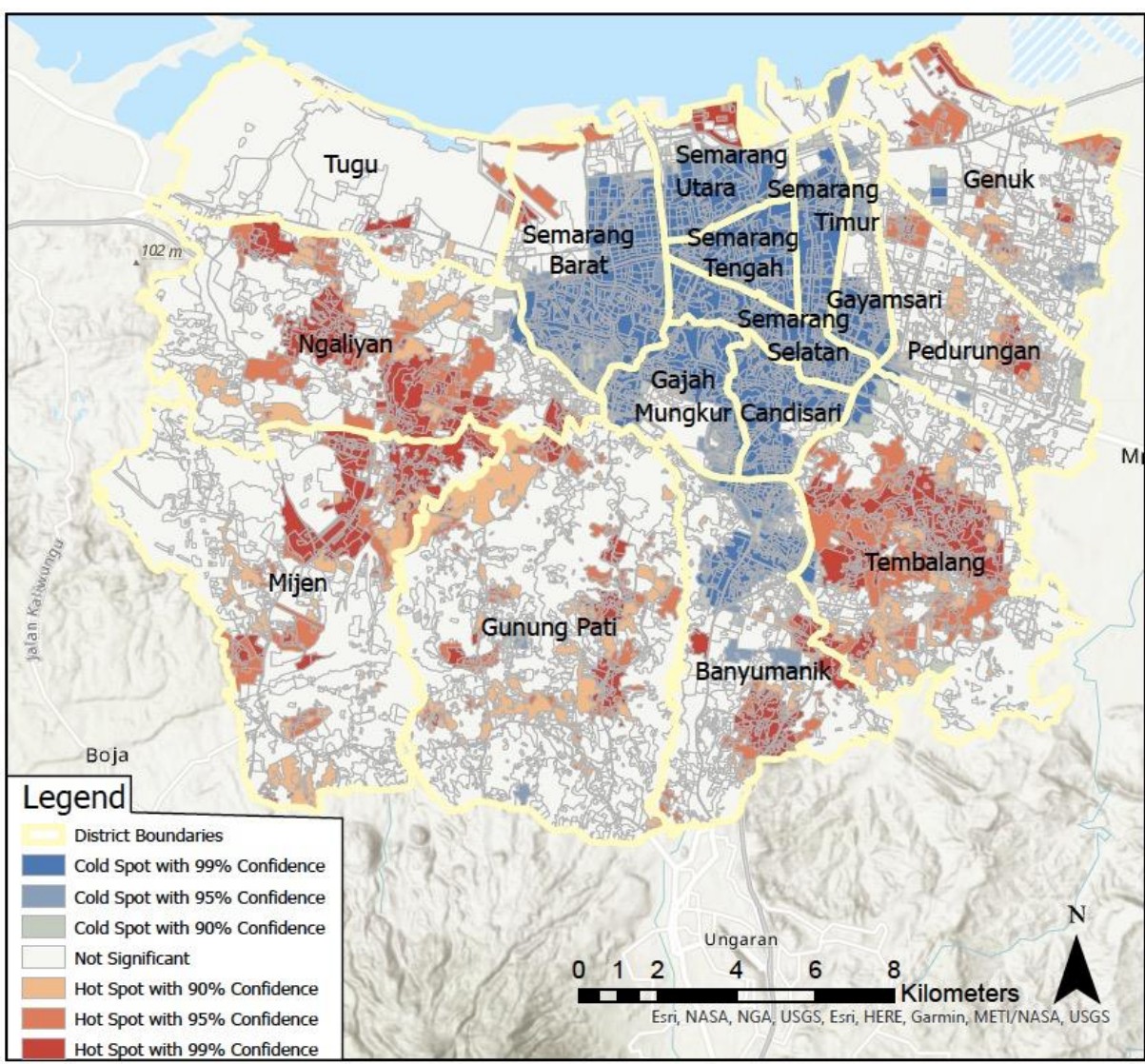

**Figure 5.** Optimized Hot Spot Analysis of areas likely to experience change in Semarang City, Indonesia. Cold spots can be seen close to the existing city center with hotspots predominantly in neighboring areas experiencing population growth and economic development.

In 2006, many areas that experienced change were predominantly agriculture or forest (Figure 6). In the 2015 data, there is an apparent loss in agricultural land; however, many new areas are classified as forests. Google Earth Pro imagery and expert analysis confirmed that these areas transitioned from agriculture to forest in those years. However, this is not a return to nature but an increase in rubber plantations. Rubber plantations consist of trees but lack the soil composition and biodiversity of flora and fauna found in forests untouched by humans [61]. Because these data appear to be based more firmly on the land cover rather than land use, this distinction between forests and plantations is not well differentiated in the ESA dataset.

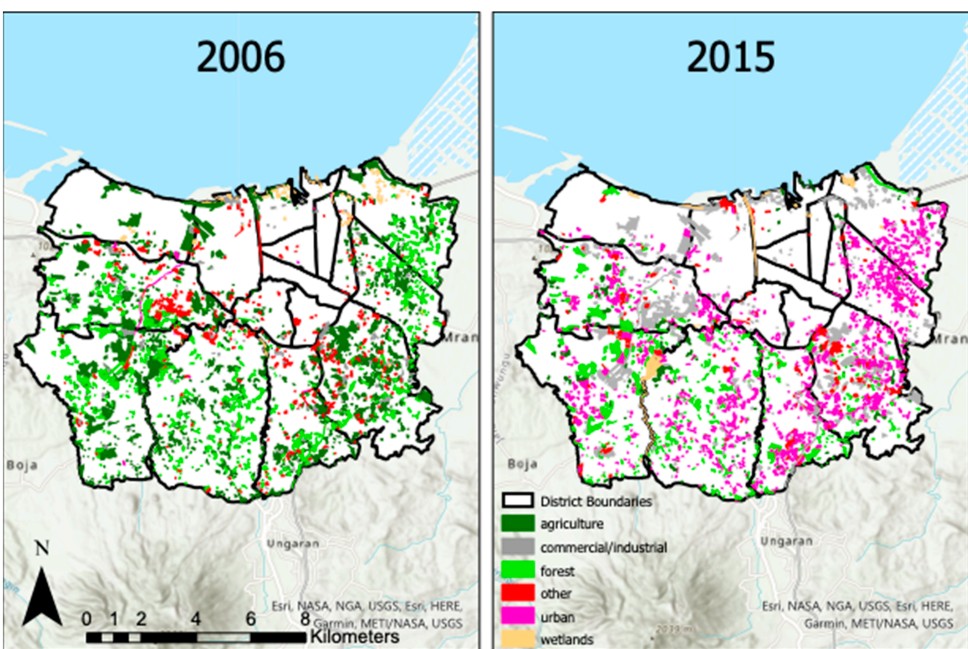

**Figure 6.** Areas of LULCC in Semarang; classes for each of the two years are highlighted. Of note are the new forested areas in 2015, the lost forest and agriculture areas from 2006, and the new urban and commercial areas in 2015.

Many of these new rubber plantations appear in the Mijen district, the district in Semarang with the highest number of farmers [62]. Mijen is approximately 53.85 km$^2$, making it the second largest district in Semarang (Table 3). In 2006, 18.65 km$^2$ was agricultural land, roughly 35% of the area. Between 2006 and 2015, a third of that land was converted from agricultural land with more than half of that (3.80 km$^2$) being converted into forest. Throughout Semarang, which has an area of approximately 384.75 km$^2$, 31.86 km$^2$ of agricultural land was converted in 2006. Thus, approximately 8% of all land in Semarang was converted from agriculture, while only 10.10 km$^2$ of that 31.86 km$^2$ was converted from agricultural land to forest. Hence, the 3.80 km$^2$ change in Mijen is relatively high for its area size. In conclusion, even though there is not a higher rate of agricultural land being transformed to another class in Mijen, there is a much higher rate of agricultural land being transitioned to forest than in other parts of Semarang (7.63% for Mijen vs. 2.63% for Semarang).

Tembalang District also had many agricultural areas (roughly 6.47 km$^2$) in 2006 which did not exist as agriculture in 2015. However, most of the land in this area was changed into commercial/industrial (2.67 km$^2$) or urban classes (2.20 km$^2$), at 6.44% and 5.30%, respectively, of the total land area in Tembalang. As in Mijen, this was a higher rate of change than was seen in the city at large. In all of Semarang, there was a total of 16.11 km$^2$ (4.19% total area) changes to commercial/industrial and 20.13 km$^2$ (5.23% total area) to urban regardless of original class in 2006.

Between 2006 and 2015, Semarang saw a decrease of roughly 33.05 km$^2$ in permeable surfaces. That is equivalent to about 10% of the land cover in all of Semarang. In 2015, the majority of the city—approximately 65.83%—was covered in impermeable surfaces.

As mentioned in the Methods section, the ESA dataset helped explain what the LULC was in 2006 and 2015 but not what had occurred in these areas between 2006 and 2015. The GEE CCDC helps explain what happened in the interim. The areas that had LULCC according to the GIS dataset experienced a range from one to four changes between 2006 and 2015 (Figure 7).

**Table 3.** Area change (km$^2$) between 2006 and 2015; percentage represent percent of overall land in Semarang that experienced that change class; columns reference class in 2015 and rows represent class in 2006.

| 2015 → 2006 ↓ | Agriculture | Forest | Commercial/Industrial | Other | Urban | Wetland |
|---|---|---|---|---|---|---|
| agriculture | 73.04 18.99% | 10.10 2.63% | 9.22 2.40% | 2.36 0.61% | 8.63 2.24% | 1.55 0.40% |
| forest | 2.22 0.58% | 80.32 20.88% | 2.97 0.77% | 0.80 0.21% | 7.84 2.04% | 0.15 0.04% |
| commercial/industrial | 0.15 0.04% | 0.19 0.05% | 44.08 11.46% | 0.68 0.18% | 1.89 0.49% | 0.04 0.01% |
| other | 0.10 0.03% | 0.50 0.13% | 2.85 0.74% | 10.96 2.85% | 1.71 0.44% | 0.11 0.03% |
| urban | 0.00 0.00% | 0.04 0.01% | 0.17 0.04% | 0.01 0.00% | 117.24 30.47% | 0.00 0.00% |
| wetland | 0.22 0.06% | 0.04 0.01% | 0.89 0.23% | 0.06 0.02% | 0.06 0.02% | 3.52 0.91% |

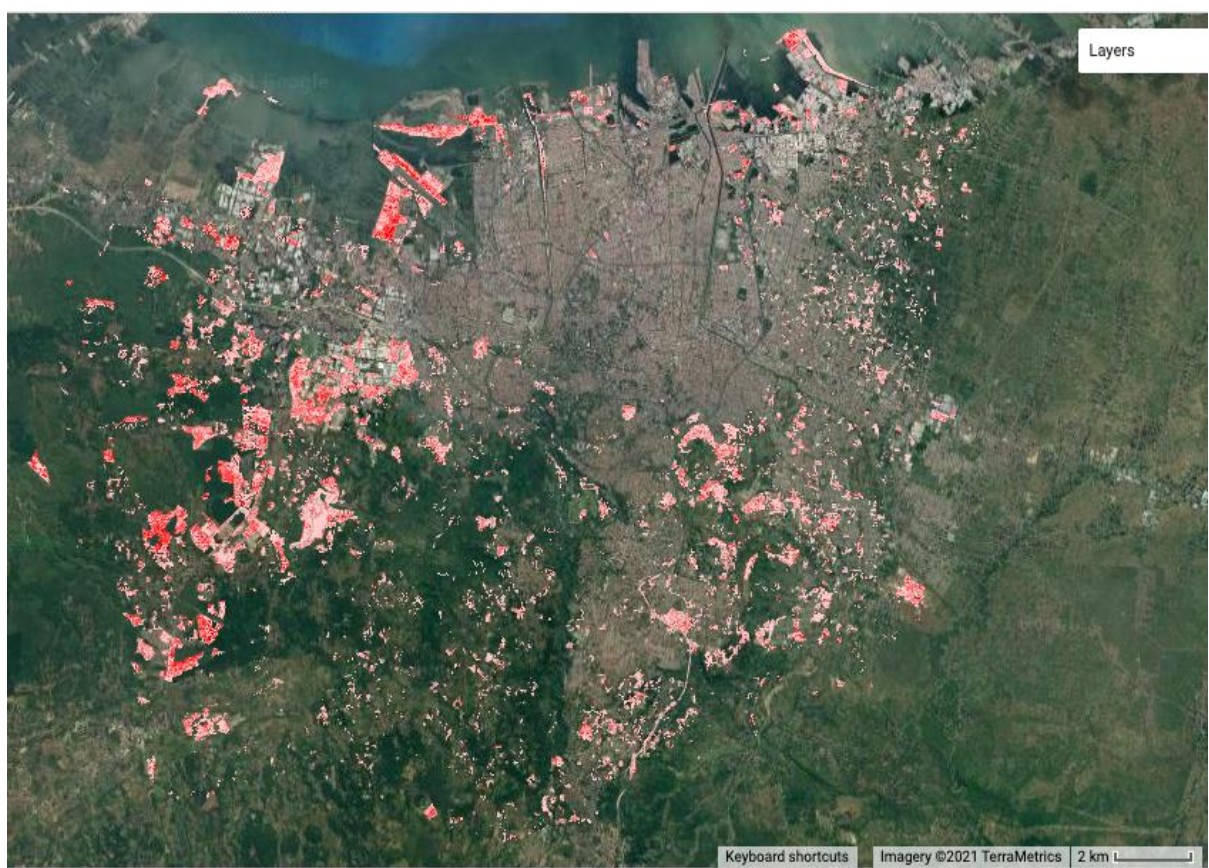

**Figure 7.** Number of Changes in areas with LULCC; darker reds indicate more changes.

One of the areas that experienced many changes is at the border of the Tugu district and Semarang Barat (West Semarang) where the Jenderal Ahmad Yani International Airport was expanded (Figure 7). CCDC detects changes in the pattern of Surface Reflectance (Near Infrared) over the course of time. Between 2006 and 2015, there were four pattern changes detected at the airport, with those changes occurring in the summer of 2007, in the spring of 2013, and in the summer of 2015 (Figure 8). In 2006, the area the airport would occupy

was classified as agriculture. Using Landsat imagery, the land can be seen to be changing from aquaculture to dried land to construction of the airport.

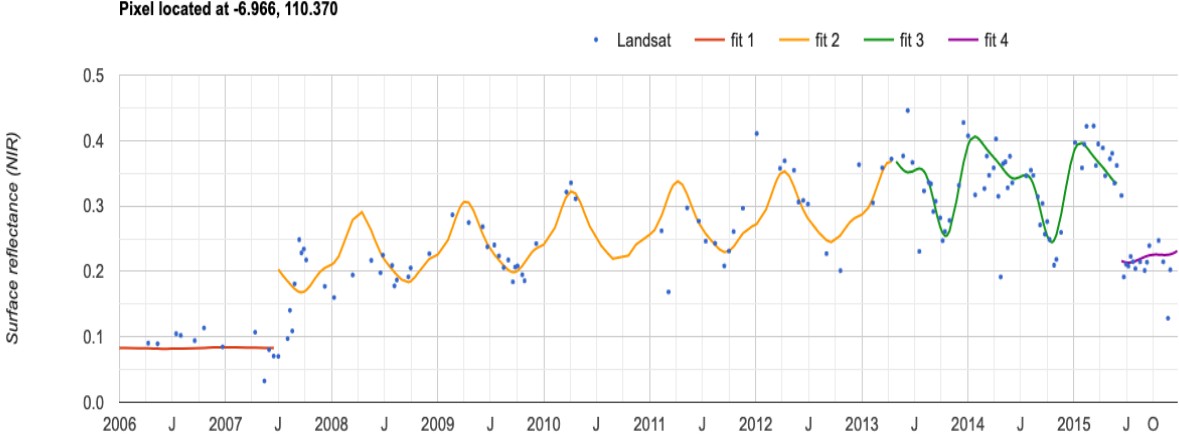

**Figure 8.** Sample CCDC of Semarang's airport.

Other areas such as in Pedurungan, which predominantly changed from forest and agriculture to urban, only experienced one or two CCDC changes during the period of 2006 to 2015. These ranged in years based on when an area transitioned from either forest or agriculture to the homes, offices, and other buildings of the urban fabric (Figure 9).

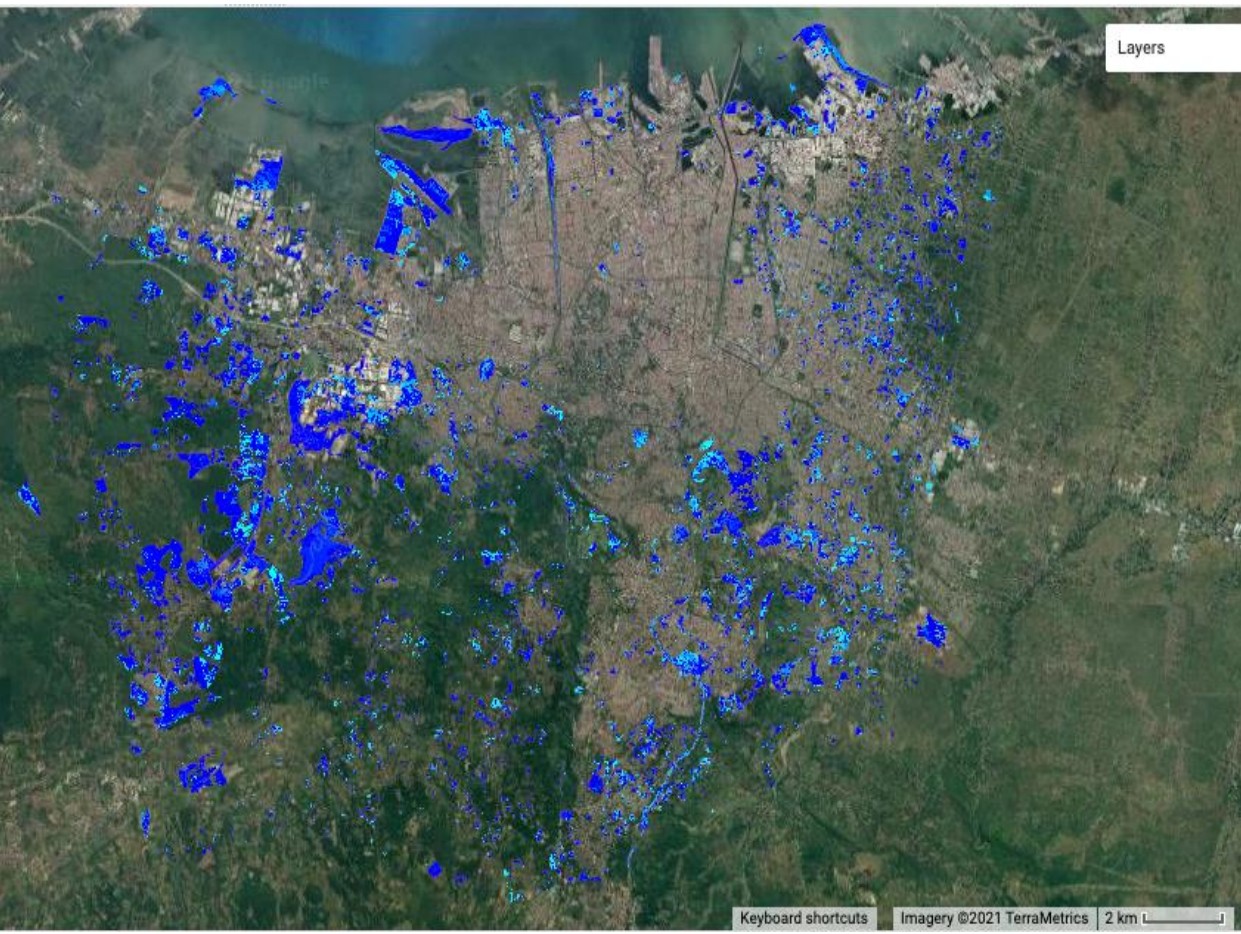

**Figure 9.** Year of Last Change in LULCC Areas; darker blues indicate more recent years (2014, 2015).

Areas in Mijen converted from agriculture to forest also experienced a higher number of CCDC changes. Such changes are indicative of the planting and growth of rubber trees and what appears to be a change in the type of agriculture during the growing season.

By observing sample areas transformed from agriculture to forest, we could determine that most of these forests were probably planted between 2008 and 2011. Using Google Maps Street View, it became clear that the rubber plantations are a mix of Rubber Agroforestry (RAF) and traditional monoculture (Figure 10). In the example from Figure 10, the RAF image shows a banana tree on the left side and another crop as a groundcover. These rubber plantations are on the younger side, as seen by the small trees and the lack of sap harvesting marks on the trunks (Figure 10). Additionally, they each have heavy undergrowth (Figure 10).

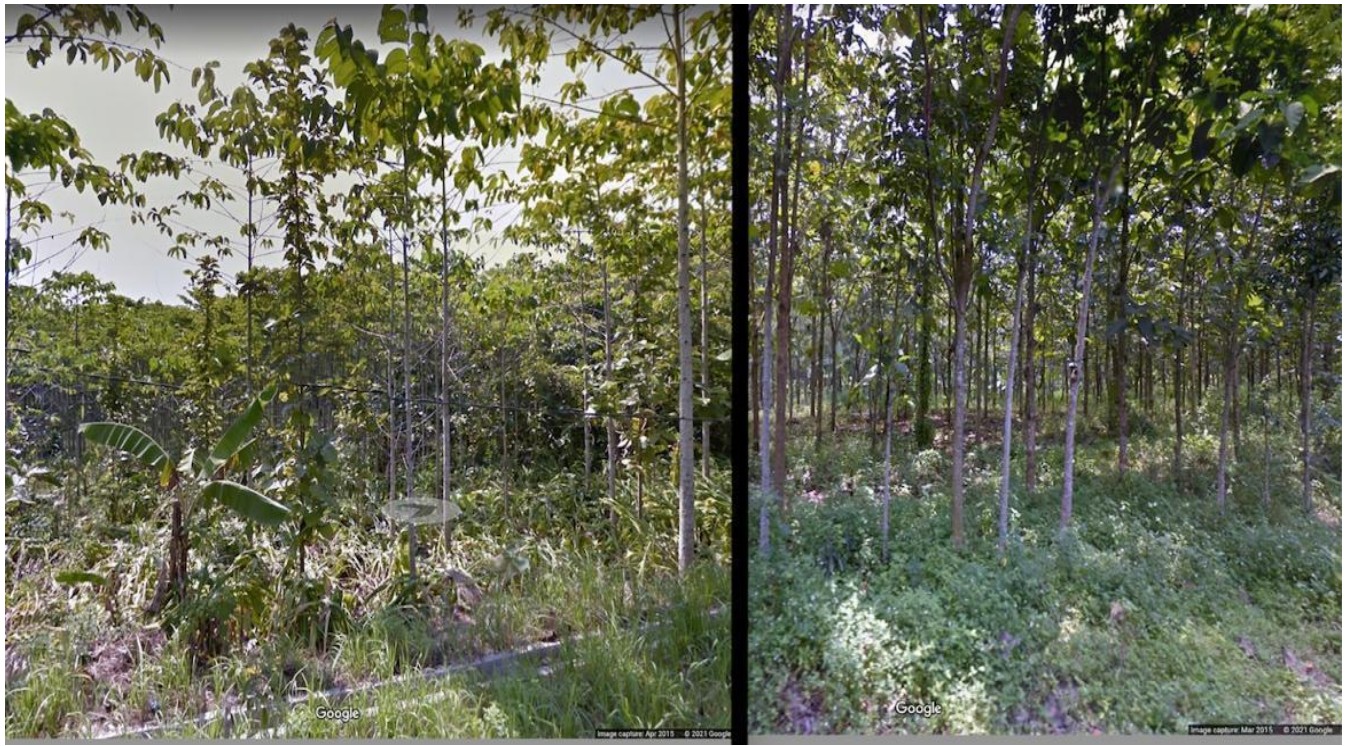

**Figure 10.** Sample of Rubber Agroforestry (left; location: 7°01′55.5″ S 110°24′23.6″ E) and Rubber Monoculture (right; location: 7°01′12.9″ S 110°19′49.5″ E); images from Google Map Street View collected April 2015 and March 2015, respectively.

## 4. Discussion

Between 2006 and 2015, there were many LULLC in Semarang City, Indonesia. In this period of time, Semarang lost 31.85 km$^2$ of agricultural land and 14.00 km$^2$ of forest land and gained 20.13 km$^2$ of urban land and 16.11 km$^2$ of commercial/industrial land. These changes are indicative of economic growth and increased infrastructure. However, there was also a loss of 33.05 km$^2$ of permeable sediments.

### 4.1. Implication of Increased Rubber Plantations

A sixth of the agricultural land in Mijen in 2006 was converted to forest lands by 2015. Of these new "forests", the majority are rubber plantations. This incredibly high turnover indicates an economic change. Rubber is a cash crop that also offers farmers the opportunity for sustenance farming in the understory. This change in Semarang is indicative of a larger transition in Indonesia. Agroforestry has been practiced since 3000 BC in Java [63], involving the cultivation and conservation of trees in agricultural practices. Prior studies show that agroforestry systems deliver carbon stock and sequestration as well as improve

soil fertility and food security [64,65]. A prior study indicates that agroforestry can store carbon ranging from 37.7 Mg ha$^{-1}$ at 1–10 years and 72.6 Mg ha$^{-1}$ at 11–30 years [66].

Historically, Indonesian farming involved a combination of traditional terrestrial agriculture and aquaculture in tidal waters and mangrove forests. However, this region is now seeing an increase in commercial agriculture, particularly rubber monocultures and Rubber Agroforestry (RAF) [61]. RAFs have the added benefit of not only producing a cash crop—rubber—but also providing food for local farmers by growing fruit trees, such as bananas, or crops, such as highland rice, beneath the rubber's canopy [61]. Although this results in slower growth of the rubber trees, and thus an increased wait time before harvest, the benefits of RAFs cannot be ignored [61].

The changes in Mijen, from traditional agriculture to rubber forests, are an indicator of an increase in economic opportunity for individual farmers in this region. As Mijen has the largest number of farmers of any district, the increase in rubber plantations, a profitable cash crop, may mean an increase in wages and better livelihood for the people of Mijen [62]. Furthermore, the presence of Rubber Agroforestry (RAF) in Meijin suggests increased sustainability, not just for the flora and fauna of the forest but for the people themselves. As aforementioned, RAF is not solely the practice of maintaining a monoculture of rubber, but also of planting other trees and vegetation. Thus, in addition to the economic benefit of money, there is also the added benefit of providing local produce to farmers and communities.

Additionally, rubber is an economically important export to Indonesia. This increase in economic growth often shadows an increase in welfare throughout developing nations [67]. The increase in rubber plantations in Semarang not only benefits farmers but also Indonesians and Indonesia as a whole.

However, rubber plantations do have downsides, particularly in regard to ecology and biodiversity. Indonesia is a land full of nutrient-rich volcanic soil [67]. However, agriculture, particularly monoculture, is a nutrient sink that would remove nitrogen and other important compounds from the earth [68]. Additionally, rubber plantations have a high likelihood of failure from climate disasters such as droughts and typhoons [68]. Finally, the clearing of land to create rubber plantations leads to the loss of conservation-critical biodiverse areas [68]. This may be less relevant in areas where these rubber plantations have been planted in lieu of agriculture, but our model does not account for areas that have transitioned from "natural" forest to rubber plantation. It is likely that this transition did occur and thus should be accounted for in some measure in this study.

The consequences of increasing rubber plantations bridge the controversy of the UN SDGs. The increased profits from rubber plantations benefit both farmers and the country as a whole; the latter fulfills the SDG call for increased GDP for developing nations (8.1) and the former builds resiliency among the poor (1.5) [38]. Previous studies have corroborated these results and suggest that in addition to resiliency, rubber plantations may also offer a reduction in inequalities due to the increase in economic growth (10.1) [38,51]. Additionally, in the case of RAFs, local farmers can both increase "agricultural productivity and incomes of small-scale food producers" (2.3) and "ensure sustainable food production systems" (2.4) [38]. However, the loss of biodiversity and the degradation brought about by the loss of forests and increase in rubber plantations directly go against the UN SDG (15.5) [38,51]. These goals are at odds with one another and thus require closer examination.

### 4.2. Spread of Urban Sprawl

In Tembalang district, there was a transition from agriculture to commercial, industrial, and urban areas. The uptick in suburban housing and economy in Tembalang is an indicator of how the urban planning of Semarang as a whole is developing. Semarang's population has burgeoned in recent years [42]. With this increase in population, there needs to be an increase in economic opportunities and safe, affordable housing for these individuals. These align with guidelines eight and nine of the United Nations Sustainable Development

Goals, which highlight the need for decent work and economic growth and industry, innovation, and infrastructure, respectively [42].

The spread of suburbia does not just mark an increase in economic growth and infrastructure but also a decrease in ecosystem services. Urban sprawl is often not the same as population growth, meaning that frequently more areas are developed than is absolutely necessary [69]. The resulting loss of ecosystem services directly violates the UN SDGs, which call not only for protecting ecosystems but for making them even stronger and more prevalent (15) [38].

Additionally, the loss of agriculture, forest, wetlands, and other greenspaces means the loss of permeable surfaces. Semarang City is prone to subsidence due to an increase in the weight of the new infrastructure and the extraction of groundwater from the aquifer [70]. The loss of permeable surfaces means that the aquifer cannot replenish on its own [70]. Thus, local people, who obtain most of their water from wells, may lose access to clean water as a result of the decrease in permeable surfaces and increase in urban and commercial/industrial zones. This directly violates the sixth SDG of the UN—namely, that people have access to clean water and sanitation [38].

### 4.3. Relevance of Processed LULC Dataset

There are several issues with this LULC ESA dataset, including that it was only run in two years and the lack of differentiation between agriculture and aquaculture. Aquaculture is very prevalent in Semarang City, especially in coastal regions of the city. It would have been interesting to see if, as a result of subsidence and saltwater encroachment, more inland areas that were previously agriculture became aquaculture. Additionally, with more classification, we could observe if these inundated areas of aquaculture transitioned to other types, since they may no longer be desirable for farming of any kind.

The presence of only two years of data means that there is an inability to validate. That is, do our expectations and findings of what occurred between 2006 and 2015 apply into the future? Granted, it has not yet been nine years since 2015, but there is still a possibility that further expansion would be visible both in regard to urbanization and increases in rubber plantations.

A final issue with this data source was that its focus was much more heavily on land cover than on land use. We can see this in their classification of an area as a forest regardless of its use as a cash crop or as a natural green area. This influences the sustainability of Semarang City in regard to the UN SDG about life on land, which concerns not only the loss of biodiversity but also the importance of sustainably managed forests [38].

### 4.4. UN SDG Policy Possibilites

Socio-demographic changes are driving factors in land cover and land use. As populations increase, there is a demand for more housing; as they become wealthier, they demand higher-quality living conditions including safer shelters, piped water, electricity, and sanitation. Facilities for these utilities are required, and the economy of a country as a whole increases, leading to increased GDP and accelerated rates of urbanization. With the expansion of commercial and industrial land use comes a decrease in forests and agricultural and other natural and semi-natural areas. This leads in turn to decreased natural resources, fewer sustainable, local foods, and increased greenhouse-gas output, overall facilitating the developing severity of climate change in coastal urban zones (which are most at risk from climate change), sea level rise, subsidence, and extreme weather events. The inundation and saltwater intrusion will affect urban landscapes next to the shoreline, access to potable water further inland, and the nearshore aquaculture endeavors in western Semarang City.

In response, smart urban planning is needed. In order to prevent a greater loss of property and lives in coastal urban zones, limitations need to be implemented to suppress further development in these areas. Efforts need to be made to plant mangroves to protect existing aquaculture from storm surge and extreme weather events. Finally, existing natural

resources and coastal barriers must be preserved to protect the city as a whole from the changing climate and related threats.

Because Semarang is a desakota, the increasing population in Central Semarang leads to LULCC pressures in the surrounding districts. Due to the limited land area of Java, there is a need to balance the protection of biodiversity and the livelihoods of residents, ensure food security, and sustain urban growth. We found that the resulting LULC varies in districts, with Tembalang shifting towards a more urban environment accompanied by increased infrastructure, while Mijen has seen a shift towards cash crops. Although these districts are part of the same city, distinct policies aligned with the UN SDGs are necessary to address their particular development.

The population is expected to continue to increase in Indonesia and, as such, urban sprawl is likely to progress [5,25]. Tembalang is already experiencing these changes and policies are necessary to ensure the quality of the lives of individuals in these environments. Models in other parts of the country suggest that, in general, decreasing urban sprawl, specifically urban fragmentation, maintains vegetated areas and consolidates sustainable infrastructure [28,29]. If Tembalang implemented a similar policy of consolidation, there may be a similar slowing of the spread of suburbia into the lands in this district still utilized for food production. Tembalang could utilize zoning to discourage development in current agricultural areas and incentivize increasing population in urban areas by providing increased infrastructure in those areas (2.3; 9.4) [38]. Other studies have shown that in developing cities, having a more compact cityscape leads to better access to clean water while also mitigating GHG emissions [71]. Additionally, by increasing density and decreasing urban sprawl, there would also be a decline in the need for heavy transportation infrastructure, which can result in habitat fragmentation (15.5) [38].

In Mijen, there is an increase in urban areas as well; however, this is coupled with a rapidly changing agricultural economy. The rise in cash crops allows for economic growth for farmers but leads to a decrease in the available food supply. For a burgeoning population, this is not sustainable. If Mijen subsidized farming to make it more profitable than exporting rubber, or offered tax incentives to farmers who replace monoculture with agroforestry, food insecurity may decrease (2.1) [38]. Currently, Indonesia needs to import rice to keep up with demand, resulting in prices that fluctuate heavily based on the global market [72]. During times of recession, this can lead to increased food insecurity among the poor. By providing supply-side subsidies, Mijen could increase rice production while maintaining livable incomes for farmers (1.4; 8.5) [38]. Decreasing rice imports would also lead to an associated GDP increase (8.1) [38].

Both Tembalang and Mijen incurred major losses of forested land between 2009 and 2016. This loss of forest is linked to a loss of biodiversity, a decrease in carbon sequestration, and an increase in habitat fragmentation. By ceasing the removal of non-crop forest and preventing "wild" forest from being converted to rubber plantations, Semarang could halt a massive amount of deforestation and provide natural landscapes for the diverse flora and fauna of those terrestrial ecosystems (15.2) [38].

*4.5. Conflicting SDGs*

Previous studies have attempted to examine the sustainability and environmental effects that current growth may have in Semarang without using the SDGs [73–75]. These studies have acknowledged the challenges of sustainability in Semarang given its current growth and future projections and models [73,74]. In this study, we have attempted to examine the LULCC through the lens of the SDGs in order to better establish suggestions and to provide a framework for potential tradeoffs.

Since the conception of the SDGs, they have been critically analyzed and reviewed by the scientific community [76–80]. Many of the SDGs associated with eliminating poverty are statistically at odds with the overall aim of the SDGs—viz., to halt climate change [76]. In general, the more compact and efficient an urban area is, the lower the GHG emissions from transportation and other forms of infrastructure [71]. However, in a city that is

actively expanding without clear policies on sustainable development, this can be incredibly challenging. Many studies of SDG 11 focus on the existing portions of cities or suburban areas rather than the surrounding areas that are becoming part of the urban tapestry due to sprawl [81,82]. As such, these studies do not comment heavily on SDG 15. As a desakota, many of the observed LULCC were from permeable non-developed lands into an urban environment.

As described in the previous Section 4.4, there are policies that Semarang and its districts could implement to conduct more sustainable development in the future. However, these do come with tradeoffs. By prioritizing terrestrial ecosystems and preserving biodiversity (SDG 15), there would be limitations on access to food (2.1), as well as people's ability to have a reliable source of income (1.2) and a growing GDP (8.1) [38]. Perhaps the best solution is not making sweeping policy changes to Semarang but instead focusing on the particular needs of each district. By continuing to monitor changing LULC and using research to drive policy, decision makers can best serve their constituencies and create lasting changes that will service both the city and the planet.

### 4.6. Implications and Future Studies

Remote sensing and new technologies have increased access to analyzing LULCC to aid in better land use planning. This sort of analysis was previously insolvable; our ability to now create comprehensive studies spanning decades and swaths of land enables us to explore the transitions of LULCC over time as seen in our GEE CCDC analysis. Furthermore, the ability to combine GIS and remote sensing data allows not only validation but also a further depth and breadth of knowledge. These yield promising results that can be implemented not only to study Semarang but in a range of developing cities and countries. This analysis allows for a way to conclusively determine how these areas are achieving the UN SDGs.

These initiatives are potentially applicable around the globe in developing communities and in those dealing with the trials of extreme weather and climate change. There is a need to further examine the conundrum of the UN SDG conflicts. However, tools provided by remote sensing and geospatial analysis offer ways to better assess and implement policies for the future. By looking back at LULCC that have already occurred, we not only have a glimpse into the future with a business-as-usual strategy but can also see other alternatives should we choose to implement even one of the aforementioned policies.

### 5. Conclusions

Semarang is experiencing rapid LULCC, and using remote sensing and GIS analysis, we determined that most LULCC was occurring in Semarang's outlying districts like Tembalang and Mijen. These transitions included the expansion of the rubber plantation industry and an increase in urban areas in previously rural or village areas. Based on these changes, we supplied potential policy ideas that would best enable Semarang as a city on a whole to support as many SDGs as possible. Many of the SDGs are in conflict, especially for a coastal city on an island. For example, land constraints mean that preserving terrestrial ecosystems (SDG 15) is at odds with the need for Semarang to feed its burgeoning population (SDG 2). The best policies for Semarang seem to be district-specific since there are disparate LULCC occurring around the metropolitan area. Policies in Tembalang should focus on discouraging urban sprawl, while Mijen policies encourage the production of food crops, such as rice, in areas already utilizing farming. Significant district-level policies should focus on urban sustainability planning and careful balancing of built urban and green spaces to constrain urban sprawl. Social vulnerability in each district must be monitored to assess physical exposure to climate change. Future studies are necessary to monitor the continuous changes in Semarang and to provide further suggestions to policymakers.

**Author Contributions:** M.K.-F. analyzed the findings and assisted in writing this paper. S.G. provided conceptual framework and discussion around SDGs and assisted in writing this paper. M.K. provided the World Bank data and critical feedback. H.P.K., M.H. and D.K. provided valuable feedback and validated the findings. L.K. reviewed the paper. All authors have read and agreed to the published version of the manuscript.

**Funding:** This research is based upon work supported by the National Science Foundation under Grant No. NSF-OISE 1827024: IRES Track I: Collaborative Research: U.S.-Indonesian Research Experience for Students on Sustainable Adaptation of Coastal Areas to Environmental Change.

**Institutional Review Board Statement:** Not applicable.

**Informed Consent Statement:** Not applicable.

**Data Availability Statement:** Not applicable since it uses source data from World Bank. The CCDC algorithm and data product mentioned in this article is available at: https://code.earthengine.google.com/8b80b591aaab91c4b20f6d5308aefb6a (accessed on 2 March 2022).

**Conflicts of Interest:** The authors declare no conflict of interest.

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
