# Peer review of "Analysis of Land Use and Land Cover Changes through the Lens of SDGs in Semarang, Indonesia"

_sustainability, doi:10.3390/su14137592_

Round 1

Reviewer 1 Report

The topic is interesting from the point of view of sustainable urban planning, sustainability and land cover.

The study has good potential but the manuscript needs some structural corrections.

Specific comments:

  1. I suggest improving the graphic part of the manuscript. Authors should make similar figures in terms of graphics, color, and form throughout the manuscript. This applies especially to Fig. 1, Fig. 2 (additionally the inscriptions are illegible), Fig. 4, Fig. 5
  2. I suggest adjusting all tables to the journal format instead of PDF / JPG tables. Similar with the formula (Line 175)
  3. The introduction lacks information about the global significance of the presented research results. I mean, how the international audience could benefit from the solutions presented by you.
  4. The discussion definitely lacks a comparison to other studies in this topic and indications of references.

Reviewer 2 Report

  • Line 13: there are 17 SDGs. Please specify the related SDGs.
  • There is no link between urban planning mentioned in the title and the content of the introduction section. 
  • Lines 25-40: please provide some citations to the information provided
  • Lines 91-93: it is better to indicate how wealthy the people of Semarang have become with a timeframe 
  • Section 2: Please justify all methodological choices (e.g., equations, techniques, analyses, etc.) and cite similar studies that used them. 
  • Section 2: Please justify the study duration (2006-2015). Why not use more recent years (e.g., 2020 or 2021)?
  • Lines 105-106: why use ArcGIS Pro for analysis if QGIS or similar GIS software can suffice? Please justify that decision.
  • Section 2.3: Also, justify the number of cases selected and the number of accuracy assessments and the techniques used, and the reclassifications done
  • Lines 136: labeling of Figure 2 is not clear. Please improve the quality of the image.
  • Section 3: all area changes are presented in square km. Why not add the percentages to understand the extent of the LULCC better
  • Section 4: Please discuss the explanations for the likely patterns of LULCC among the various districts.
  • The discussion section should relate findings to the study objective and underscore the value the study adds to the literature and whether the method can be applied broadly outside the study area.
  • It should also discuss how the findings support or differ from similar LULCC studies in other cities in Indonesia. 
  • Section 5: the conclusion section state the importance of the findings to the field, including implications for urban planning policy and practice.

Reviewer 3 Report

Dear authors,

The manuscript presents an important analysis of sustainable urban planning in Semarang, Indonesia. The article fits the theme of the scope and is unpublished for the study region. It is an interesting manuscript, but the introduction, figures and description of the methodology must be improved. It presents a case study and, as far as I can be competent to judge, it is a good paper. Next, I place my suggestions:

Lines 24-102: This section is without citations, it is necessary that you have them as it is an introduction. Only as an example, The first paragraph has several affirmations and lacks appointments. The introduction is poor in extension, development and appointments. I suggest expanding this section, a good approach is to place a paragraph at the beginning the general context more context related to Land Use Change. For example, perform the overview of the effects on LULC in different parts of the world, associated with agricultural expansion, urban expansion and engineering projects such as access, energy, among others. I place several manuscripts that can help develop this: Agricultural expansion (https://doi.org/10.1073/pnas.0606377103, https://doi.org/10.1073/pnas.1111374109), Urban expansion (https://doi.org/10.1016/j.scitotenv.2017.12.143), Engineering projects (https://doi.org/10.3390/ijgi9100583, https://doi.org/10.1016/j.landusepol.2013.01.009, https://doi.org/10.3390/ijgi10030191).

Lines 97-99: The goal of the paper is not clear.

Lines 103-120: This section must be improved, I suggest placing as a paragraph and not listed as it is. It is necessary to place a map that shows the study area, with the countries around the study area. This thinking more of your non- Indonesia readers, who may not be very familiar with the territory of Asia.

Line 136, 215, 228, 262, 278: Figure 2, 4, 5, 6 and 8 needs to be improved. The quality of the image is terrible, it is impossible to appreciate details. Avoid the use of blue color to represent territories, it is more common to use it for bodies of water (oceans, rivers, lakes). The scale of colors data is not adequate. Labels cannot be read, they should be improved

Line 148, 159, 246: The tables do not meet the format required by the journal.

Lines 394-461: This section should be synthesized. Comment exclusively as relevant. This section must respond to the objective of the paper, since the objective is not clear, I do not see the conclusions very clearly.

Line 475-540: The paper has few references, they are just 29. It is an insufficient number for a journal article.

Round 2

Reviewer 2 Report

The paper has been improved substantially. Please address the following remaining issues:

- There is redundancy in the use of Indonesia (used over 80 times throughout the manuscript). Please use a pronoun in some places instead.

- The material and methods section begins with Figure. Please describe a figure above and not beneath it. 

- There is confusion on page 7 about which one is figure 2. The one below is not legible, while the one above lacks a title

- In the conclusion, the authors should list the important district-level policies that the authors think are best for the study area.

Reviewer 3 Report

Dear authors,

This is the second review of the manuscript, the authors attended to all the suggestions made. It presents a case study. 

Author Response

Thank you for your feedback!